# Application of the Human Proteome in Disease, Diagnosis, and Translation into Precision Medicine: Current Status and Future Prospects

**DOI:** 10.3390/biomedicines13030681

**Published:** 2025-03-10

**Authors:** Yawen Xie, Xiaoying Chen, Maokai Xu, Xiaochun Zheng

**Affiliations:** Department of Anesthesiology, Fujian Provincial Hospital, Fuzhou 350001, China; 957995348@fjmu.edu.cn (Y.X.); ashley8361@sina.com (X.C.)

**Keywords:** human proteomics, precision medicine, biomarkers discovery, disease diagnosis, personalized treatment

## Abstract

This review summarizes the existing studies of human proteomics technology in the medical field with a focus on the development mechanism of a disease and its potential in discovering biomarkers. Through a systematic review of the relevant literature, we found the significant advantages and application scenarios of proteomics technology in disease diagnosis, drug development, and personalized treatment. However, the review also identifies the challenges facing proteomics technologies, including sample preparation of low-abundance proteins, massive amounts of data analysis, and how research results can be better used in clinical practice. Finally, this work discusses future research directions, including the development of more effective proteomics technologies, strengthening the integration of multi-source omics technologies, and promoting the application of AI in the human proteome.

## 1. Introduction

The human proteome, which is the end product of gene expression, plays a central role in the physiological and pathological processes of an organism [1,2]. It encompasses all proteins expressed in every cell, tissue, and organ of the human body, and serves as a bridge that connects the genotype and phenotype [3]. Proteins are the primary executors of biological functions and participate in nearly all life activities, including the cellular structure, metabolism, signal transduction, immune response, and more [3]. Therefore, a deep understanding of the complexity of the human proteome and its role in health and disease is highly important for advancing biomedical research.

In recent years, significant progress has been made in human proteomics research, with the rapid development of high-throughput sequencing technologies, mass spectrometry, bioinformatics, and other technologies [4,5]. These advancements have enabled large-scale, high-throughput protein identification, quantification, and functional analysis [4]. Mass spectrometry can separate and identify proteins with high resolution, whereas bioinformatics provides powerful tools for data processing and analysis [6]. Additionally, the establishment of multiple proteomic databases, such as UniProt and the Human Protein Atlas, offers researchers abundant resources and tools and facilitates the integration, sharing, and interpretation of proteomic data [7,8].

In fact, proteomics is increasingly applied in disease research, diagnostics, and precision medicine [9]. By comparing the proteomic differences between healthy and diseased states, researchers can reveal the molecular mechanisms of diseases [4,10] by discovering protein biomarkers associated with specific diseases, and proteomics technology can aid early diagnosis and prognostic evaluation [11,12]. In precision medicine, proteomics provides critical insights for personalized treatment. By understanding the proteomic characteristics of a patient, physicians can tailor more precise treatment plans [13].

This paper aims to review the current status of human proteomics applications in disease, diagnostics, and precision medicine and explore recent advancements, challenges, and future prospects. Through an in-depth analysis of the role of proteomics in biomedical research, we hope to provide valuable references and guidance for researchers in related fields.

## 2. Proteomics

### 2.1. Traditional Proteomics

Proteomics was initially defined as a technique to analyze all proteins expressed by the genome of an organism, including their structure, function, and interactions [1]. Due to the extreme complexity of the proteins expressed by the human genome, the detection process for human proteomics involves multiple steps, from extracting protein samples to data analysis, each of which requires precise operations to ensure high-quality results. Since proteins have post-translational modifications and various isomers, peptide fragments are easier to handle and separate than intact proteins, which simplifies the sample preparation process. Moreover, because peptide fragments are more easily detected by mass spectrometers, proteomic analysis typically uses a bottom-up approach [14].

The proteomics detection process can be divided into four steps: (1) acquisition of samples; (2) processing of samples; (3) mass spectrometry analysis; (4) data analysis, integration, and analysis of biological information (Figure 1). Researchers can extract protein mixtures from tissues, body fluids, or cells, perform reduction and alkylation treatments, and digest them into different peptide fragments using enzymes. The processed peptide fragments are separated by liquid chromatography (LC) and detected by mass spectrometry (MS) for their mass-to-charge ratio [15]. The data obtained from the mass spectrometer are initially analyzed, peptide fragments are subsequently identified through database searches and algorithms, and proteins are quantified to determine their abundance. Finally, differential expression and network analyses of the proteins’ biological information are conducted to obtain comprehensive proteomic information.

### 2.2. Tech Advance

With the continuous progress of technology, proteomics has evolved from the initial two-dimensional electrophoresis and mass spectrometry methods for high-throughput protein analysis, protein structure prediction, and complex protein interaction network research. By greatly increasing the resolution and sensitivity of proteomics, researchers have gained a deeper understanding of the role of proteins in various biological processes and revolutionized drug development, disease diagnosis, and systems biology. The development of these technologies accelerates the pace of scientific research and lays a solid foundation for future biomedical innovation.

#### 2.2.1. Development of the Sample Processing Technology

Proper processing of biological protein samples is a critical step in obtaining high-quality proteomics data. Accurate sample processing methods can significantly improve the sensitivity and accuracy of the MS analysis. However, this phase is often prone to problems. Traditionally, protein samples have been processed via gel technology [16], which efficiently extracts high-quality peptides from gels. However, gel cutting, especially for low-abundance proteins, may cause losses, which affects the experimental results. Recently, gel-free protein sample preparation methods have rapidly developed and show significant advantages in improving the experimental efficiency, reducing sample loss, and increasing the flux and coverage; thus, they have been widely used in modern proteomics studies. Typical gel-free protein sample preparation methods include filtration-assisted sample preparation (FASP) [17], in-solution digestion, and single-pot solid-phase enhanced sample preparation [18]. FASP is a method that processes SDS-dissolved cells in the form of a proteomic reactor [19]. It removes SDS and unbound proteins through ultrafiltration membranes and adds proteases for digestion, and the resulting peptides are purified by membrane filtration. The advantage of FASP is that it can handle relatively large amounts of protein samples and is compatible with various proteolytic enzymes, which enables sequential digestion with different proteases [20]. The SP3 technique is based on hydrophilic interaction mechanisms and uses paramagnetic beads to quickly, robustly, and efficiently process protein samples for proteomic analysis. SP3 is compatible with various chemical reagents for cell lysis and protein solubilization to simplify the experimental workflow [21]. With the continuous development of single-cell proteomics, improving the recovery of proteins and peptides has become more challenging. A research team has proposed a new single-cell digestion method known as the WinO method [22]. This method does not require special equipment and improves the recovery of proteins and peptides by reducing the contact area between sample solution and plastic containers and avoiding sample adsorption onto the tips of pipettes. However, this method relies on specific liquid chromatography–mass spectrometry systems and specific analysis software in practical operation, which limits the application of the technique.

#### 2.2.2. Development of MS

After years of development, mass spectrometry in proteomics has gradually become a core tool to study the structure and function of complex proteins. Mass spectrometry technology was first applied to organic chemistry and isotope studies. In the early 20th century, physicists Francis Aston and J.J. Thomsen developed the initial mass spectrometer, which laid the foundation for the development of mass spectrometry. However, early mass spectrometry techniques were mainly used for small-molecule analysis and have not been applied to study proteins or biological macromolecules. In the 1980s, two key ionization techniques were successfully invented: electrospray ionization (ESI) [23] and matrix-assisted laser desorption/ionization (MALDI) [24]. These two technologies have enabled the ionization of macromolecular proteins in mass spectrometers and pioneered the application of mass spectrometry in proteomics. In the 2000s, tandem mass spectrometry (MS/MS) was widely used. Tandem mass spectrometry can complete the molecular weight determination and peptide sequence determination in one analysis, which significantly improves the accuracy and coverage of protein recognition. Liquid chromatography–mass spectrometry (LC–MS/MS) has also gradually become the mainstream method of proteomics research and greatly improved the flux of samples and analytical ability of complex protein mixtures [25,26,27]. To quantitatively analyze protein expression levels, several labelling and label-free quantification techniques have been developed, such as relative and absolute quantification of stable isotope labelling (SILAC) [28,29] and label-free quantification (LFQ) [30,31]. The introduction of these technologies enables mass spectrometry to be used to identify proteins and precisely measure the expression levels of proteins. At present, mass spectrometry analysis is not limited to protein identification and quantification but extended to the study of protein post-translational modifications (e.g., phosphorylation and glycosylation). The application of multi-dimensional mass spectrometry (e.g., mass spectrometry imaging technology) and integrated multi-omics (e.g., combining genomics and metabolomics) also provides a more comprehensive understanding of protein function and interaction networks [32].

### 2.3. Possible Potential Scopes in Medicine

Proteomics technology has wide applications in the medical field, especially in the discovery and verification of disease biomarkers, and is particularly critical for the early diagnosis of diseases. For example, in a study of 1763 proteins, Hsu et al. [33] successfully identified and validated six potential cancer biomarker proteins. These studies greatly advance our understanding of cancer proteomics, although these studies were conducted in different laboratories with large differences in standards and methods. In addition, the application of proteomics is reflected in personalized medicine, which helps doctors understand the protein expression patterns of patients in detail to develop targeted treatment plans. For example, Kikuchi et al. [34] identified 3621 proteins via unlabeled shotgun proteomic analysis of non-small-cell lung cancer 21 and reported that the PAK 2 protein was significantly upregulated in non-small-cell lung cancer samples, which suggested that PAK 2 was a potential biomarker and a drug target. Experiments have shown that the use of the PAK 2 inhibitor IPA-3 can effectively inhibit the migration and invasion of non-small-cell lung cancer cells, which provides a new idea for the treatment of this species [34]. These results demonstrate the potential of proteomics in disease diagnosis and individualized treatment and highlight the importance of standardizing and unifying methodological approaches. The application potential of proteomics in the field of anesthesia is also significant. In clinical practice, anesthesiologists still administer anesthetic drugs with unclear mechanisms of action based on the drug distribution curve and observe the response and vital signs of patients to increase the dose. Proteomics can help understand the specific mechanism of action of anesthetic drugs, help identify biomarkers related to the anesthetic drug response, and help doctors develop individualized anesthesia plans for each patient to reduce the occurrence of adverse reactions [35].

## 3. Application of Proteomics in Disease Research

Proteomics plays an important role in disease research and clinical applications. By analyzing protein differences in disease versus healthy states, proteomics can help scientists and doctors gain a deeper understanding of the biological mechanisms of disease. This understanding includes identifying protein changes associated with specific diseases that may serve as novel biomarkers for the early diagnosis of disease, prognostic assessment, or treatment response monitoring.

### 3.1. Research on Disease Mechanisms

Abnormal expressions of proteins, functional changes, post-translational modifications, and the reconstruction of protein interaction networks are closely related to the occurrence and development of diseases. Many diseases, including neurodegenerative diseases and cancer, involve disorders of the proteome. Alzheimer’s disease (AD) is an age-related neurodegenerative disease and the most common form of dementia [15]. The deposition of insoluble protein aggregates is a hallmark of neurodegenerative diseases. Bai et al. [36] reported the most comprehensive analysis of the brain-insoluble proteome, which covered 4216 proteins in AD, including 36 differentially expressed proteins. As hypothesized, the most enriched proteins are A β, tau, APOE and complement components, and proteins in the RNA splicing, phosphorylation regulation, synaptic plasticity and mitochondrial function. This study confirmed that the abnormal accumulation of Aβ amyloid was one of the pathological features of AD. A further comparison of RNA from the AD group and control brains showed that the accumulation of unspliced RNA species in AD caused dysregulated RNA processing, including myc box-dependent interacting protein-1, aggregated protein, and presenilin-1. Thus, with aberrant RNA, splicing is involved in AD pathogenesis (Figure 2). The application of multi-dimensional mass spectrometry (e.g., mass spectrometry imaging) and integrated multi-omics (e.g., combining genomics and metabolomics) is mainstream in proteomics. Studies have used proteome–transcriptome comparisons to show that there are RNA-dependent and RNA-independent expression changes in AD [37,38]. Strikingly, these RNA-independent DE proteins, such as MDK, PTN, NTN 1, SMOC 1, SFRP 1, SLIT 2, HTRA1, and FLT 1, are generally highly associated with Aβ levels and enriched in amyloid plaques [37,39,40]. MDK, NTN 1, and SFRP 1 are directly bound to the Aβ peptide [37,38,39,40]. The occurrence and development of cancer are also closely related to abnormalities in proteins. Proteomic techniques can identify tumor-specific proteins and reveal key signaling pathways and biological processes involved in tumorigenesis. For example, certain oncogenes encode proteins are highly expressed in tumors, whereas proteins encoded by tumor suppressor genes may be inactivated or poorly expressed. Zhang et al. [41] analyzed 174 TCGA human ovarian cancer samples using label-free shotgun LC–MS/MS and identified 9600 proteins and 24,429 phosphorylation sites in total. They also classified tumors into subtypes based on their proteomic characteristics and identified functionally related proteins, such as P53 and BRCA 1/2 [41].

### 3.2. Discovery of Biomarkers

The exciting progress in MS-based proteomics analysis of cerebrospinal fluid (CSF) can be used to discover biomarkers of diseases in general [37,38,42,43,44]. In many proteomics experiments, high-abundance proteins often mask low-content proteins, so the immuno-elimination of these abundant proteins is a common method that improves the detection of low-abundance proteins. Sathe et al. [42] reported in-depth studies of CSF in five controls and five AD patients, where they first performed immunoremoval to remove the 14 most abundant proteins from CSF samples and subsequently analyzed them using the TMT-LC/LC–MS/mass spectrometry method. In this study, 2327 proteins were quantified with 139 DE proteins, including MAPT, NPTX 2, VGF, GFAP, NCAM 1, PKM, and YWHAG [42] (Figure 3). Higginbotham also enhanced the depth of the CSF proteome by analyzing 2875 proteins from 20 controls and 20 AD cases and revealed 528 DE proteins, including MAPT, NEFL, GAP 43, FABP3, CHI3L1, NRGN, VGF, GDI 1, and SMOC 1 [38]. The cancer-associated proteome is currently the most extensively studied proteome of human diseases. Using itraq-based LC–MS/mass spectrum, Hsu et al. [33] identified and verified six potential lung cancer-associated biomarker proteins (ERO 1 L, PABPC4, RCC 1, RPS 25, NARS, and TARS) among 1763 proteins and verified these six proteins using immunohistochemical staining and protein blot analysis. Similar studies have advanced in metabolic diseases, and a study that monitored the plasma proteome of 43 obese individuals who experienced sustained weight loss revealed individual-specific protein levels. Using a mass spectrometry-based plasma proteome analysis, the study measured 1294 plasma proteomes. Longitudinal monitoring of the cohort revealed individual-specific protein levels, where the broad impact of weight loss on the plasma proteome was reflected in 93 significantly affected proteins. Among these proteins, SERPINF1 and apolipoprotein APOF 1 are the most significantly regulated and can be used to monitor metabolic diseases [45].

### 3.3. How Proteomics Helps

#### 3.3.1. Comprehensive Analysis of Protein Expression

Proteomic techniques enable a comprehensive comparison of protein expression differences between normal and diseased tissues, which helps reveal the aberrant expression patterns of disease-associated proteins. For example, in breast cancer studies, the HER 2 protein is significantly overexpressed in cancerous tissues and closely associated with the aggression and poor prognosis of cancer [46]. Similarly, in studies of gestational diabetes mellitus (GDM), proteomic techniques such as mass spectrometry have revealed several potential urinary biomarkers such as coagulation factor IX and the trans-α-trypsin inhibitor heavy chain H4 (ITIH4). These methods can be combined with statistical methods to identify and validate these biomarkers. Through these techniques, researchers have been able to identify specific proteins and metabolites associated with GDM and provided new perspectives for the early diagnosis of GDM [47].

#### 3.3.2. Changes in Protein Modifications

Post-translational modifications of specific proteins, such as phosphorylation and acetylation, are associated with many diseases. Phosphorylation plays a key role in signaling and is closely related to the progression of cancer [48]. Using mass spectrometry, researchers can enrich and detect phosphopeptides, identify specific phosphorylation sites, and reveal aberrant signaling pathways such as the *PI3K/AKT* and *MAPK* pathways. In addition, ubiquitination is a key modification associated with neurodegenerative diseases, infectious diseases, and tumorigenesis. By detecting ubiquitinated peptides via mass spectrometry and combining genomic and transcriptomic data, researchers can identify potential therapeutic targets and biomarkers [49].

#### 3.3.3. Study of Protein–Protein Interactions

Proteomics technology can map protein interaction networks to help identify potential disease driver proteins or pathways. Vinayagam et al. [50] combined affinity purification mass spectrometry (AP-MS), RNA interference screening, and quantitative phosphoproteomics to monitor the response of the insulin receptor/*PI3K/Akt* network to insulin signaling and conducted a comprehensive analysis of the insulin signaling network in fruit flies. Researchers used AP-MS technology to map a protein–protein interaction (PPI) network with approximately 20 core members of the insulin signaling pathway and characterized the functions of these interacting proteins using RNA interference (RNAi) technology. Additionally, researchers identified target proteins on the signaling pathway using phosphoproteomic data. They identified protein complexes that were stably associated or dynamically assembled with the insulin signaling pathway and classified these complexes. Researchers also used the Protein Complex Enrichment Analysis Tool to organize protein complexes in the network and analyzed their relationships with the insulin signaling pathway, such as activation or inhibition. Overall, this study provides a comprehensive resource on the insulin signaling network using proteomics technology and offers significant data support to understand the structure and function of the insulin signaling pathway [50]. Traditional AP-MS methods are very flexible but may not detect weak interactions mediated by short linear motifs (SLiMs). Peptide-based interaction proteomics technology can overcome this limitation. The intrinsically disordered regions or short linear motifs (SLiMs) were identified, and short peptide sequences covered these regions. Mass spectrometry was used to identify proteins that bound to the peptide and recognize and quantify proteins that interacted with specific sequences or modifications [51].

## 4. Application of the Human Proteome in Precision Medicine

### 4.1. Diagnosis

Proteomics technology has been used to discover protein biomarkers associated with cardiovascular diseases, cancers, and neurodegenerative diseases, which increases the accuracy of early diagnosis, promotes early intervention, and improves global health levels. This early prevention and diagnosis technology is particularly important in countries with limited resources, since it can alleviate long-term medical burdens. Proteomic technology has made progress in the diagnosis of ischaemic stroke. By analyzing the proteomes of the brain and cerebrospinal fluid after stroke using high-throughput techniques, researchers have identified several candidate biomarkers that can be used for stroke management and may also serve as targets for treatment, such as the glial fibrillary acidic protein (GFAP) and retinol-binding protein 4 (RBP 4). The expression levels of these two proteins differ between intracerebral haemorrhage (ICH) and ischaemic stroke, which can be used to distinguish subtypes of stroke [19]. In terms of cancer, a research team first used immunopeptidomics technology to discover that CT45 was a naturally occurring cancer antigen and presented as an HLA I receptor on ovarian cancer cells, and HLA I peptides derived from CT45 promoted the ability of CTLs to kill tumor cells in patients. An analysis of the proteome of ovarian cancer cells treated with DNA demethylation drugs revealed that CT45 was one of the most upregulated proteins. These findings indicate that CT45 can be a target for ovarian cancer immunotherapy and a promising candidate biomarker [52]. Proteomes and subproteomes can be characterized within cells at single-cell resolution by combining imaging and high-resolution mass spectrometry [53,54]. This approach has been used to study the subproteome of endosomal vesicle transport [55]. Since these endolysosomal networks are dysregulated in neurodegenerative diseases, the characteristics of potential biomarkers can be detected [56]. Proteomic analysis of post-mortem spinal cord tissue using LC–MS/MS revealed that ATP5D and calmodulin were downregulated in sporadic ALS patients compared with the control individuals [57].

### 4.2. Personalized Treatment

Personalized treatment is one of the core concepts of precision medicine. Through proteomics, we can gain a deep understanding of the molecular mechanisms of individual diseases and design targeted treatment plans. Currently, the most widely studied proteome is the cancer proteome, and research on breast cancer is one of the hot topics. Studies have confirmed the main molecular subtypes of breast cancer: Luminal A, Luminal B, HER2-positive, and Basal-like subtypes [58]. Each subtype shows unique gene expression and protein expression profiles, which suggests different biological behaviors and treatment responses. In 2012, a research team used RPPA technology to perform protein analysis on breast cancer samples. Data analysis confirmed at least two clinically defined HER2-positive breast cancer subtypes: the HER2E-mRNA subtype and the luminal-mRNA subtype. HER2E-mRNA subtype/HER2-positive tumors presented increased expression of *HER1/EGFR, p-HER1, HER2, p-HER2, p-SRC* and *p-S6*, whereas these features were not obvious in luminal-mRNA subtype/HER2-positive tumors; in contrast, luminal-mRNA subtype/HER2-positive tumors presented increased expression of protein markers associated with luminal cancer, such as GATA3, BCL2 and ESR1. In addition, this study combined a genetic analysis to reveal multiple potential therapeutic targets in HER2-positive breast cancer. For example, in the HER2E-mRNA subtype, researchers reported a relatively high frequency of *PIK3CA* mutations, low-frequency mutations in *PTEN* and *PIK3R1*, and gene loss of *PTEN* and *INPP4B* [59]. Proteomics can be used to distinguish different molecular subtypes of HER2-positive breast cancer in clinical practice and reveal differences in signal transduction pathways, protein expression, and potential therapeutic targets among these subtypes. These findings deepen our understanding of HER2-positive breast cancer and provide a scientific basis for more precise individualized treatment strategies, especially HER2-targeted treatment. Currently, many studies integrate proteomics with other technologies and can be used to predict the responsiveness of patients to treatment, guide individualized treatment, and optimize it. *CDK4/6* inhibitors combined with endocrine therapy (ET) are the standard first-line treatment for patients with HR+/HER2− advanced breast cancer (without visceral crisis) [60]. However, some patients rapidly experience disease progression after the first-line treatment, and there is currently no standard regimen recommended after progression on first-line treatment. One study used a proteomic technique called reverse-phase protein array (RPPA) to analyze tumor tissue samples from patients with metastatic breast cancer. The RPPA technique enables the simultaneous quantification of the expression and post-translational modification levels of multiple proteins. This method enables the comparison of biomarker expression and/or activation levels between responders and non-responders, which can help develop more personalized treatment plans for patients with HR+/HER2− advanced breast cancer [61]. Proteomic technology can also be used to predict the response of neurodegenerative diseases to treatment. Researchers analyzed personal patients’ blood or cerebrospinal fluid samples using technologies such as liquid chromatography–mass spectrometry (LC–MS) to identify metabolites associated with Parkinson’s disease progression. These metabolites may be associated with specific metabolic pathways or enzyme activities. Next, the researchers analyzed the expression levels and modification status of key proteins in these metabolic pathways. By comparing the expression patterns of metabolites and related proteins in patient samples with clinical data (such as response to dopamine replacement therapy), they reported that changes in the expression of key enzymes in the dopamine metabolism (such as tyrosine hydroxylase and DOPA decarboxylase) might affect the effect of dopamine replacement therapy. By analyzing the levels of metabolites related to these enzymes (such as dopamine precursors or metabolites), researchers can predict the response of a patient to treatment [62].

### 4.3. Drug Research and Development

#### 4.3.1. Study of Drug Mechanisms

Proteomic techniques play crucial roles in studying the mechanism of action of drugs, especially in revealing how drugs affect the protein expression, modification and interaction networks. Through mass spectrometry, researchers can analyze the global protein changes in cells or tissues before and after drug treatment in detail to understand the mechanism of drug action. A study using quantitative proteomics to analyze the effects of the *EGFR* inhibitor erlotinib on non-small cell lung cancer (NSCLC) cell lines showed that the drug significantly downregulated the phosphorylation levels of several key proteins such as p-AKT and p-mTOR, which are involved in the *PI3K/AKT/mTOR* pathway. In addition, the study identified protein markers associated with drug sensitivity and resistance, such as different phosphorylation statuses of PTEN and the upregulation of p-ERK, which suggests the activation of alternative signaling pathways [63]. Although epidermal growth factor receptor-tyrosine kinase inhibitors (EGFR-TKIs) are beneficial for treating non-small cell cancers positive with *EGFR* mutations [64,65], the acquired drug resistance remains a critical clinical problem. Nearly 50% of NSCLC patients with acquired resistance to EGFR-TKIs have a secondary mutation in *T790M* (a secondary point mutation in EGFR exon 20 replaces methionine at amino acid position 790) [66,67]. In 2014, a team identified various T70 proteins associated with the *T790M* mutation, which are considered “co-drivers” of the EGFR-TKI resistance. These factors include receptor tyrosine kinases such as MET and IGF1R, which are activated in the context of the *T790M* mutation and may play important roles in the survival and drug resistance of tumor cells [68]. Proteomic technology can be used to analyze the mechanism of drug action and to study the mechanism of drug resistance in some patients, which deepens our understanding of the molecular action of drugs and promotes the development of individualized treatment options.

#### 4.3.2. Assessment of Drug Safety

The toxicity of drugs and their safety in the clinic remain difficult problems for pharmaceutical and biotech companies, so finding predictive biomarkers of early drug toxicology can minimize adverse reactions in the later stages of drug development. In 2004, a team used two-dimensional differential gel electrophoresis and mass spectrometry to identify the proteome profile associated with hepatocyte steatosis after the administration of compounds to rats during the preclinical development phase and reported changes in the expression of many proteins that may be related to the known toxicological mechanisms of hepatic steatosis. These changes include the upregulation of pyruvate dehydrogenase, which is involved in the acetyl-CoA production, phenylalanine hydroxylase, and A-ketoisovalerate dehydrogenase, and the downregulation of sulphite oxidase, which may play a role in the triglyceride accumulation. Furthermore, the downregulation of the chaperone-like protein glucose regulator protein 78 coincides with the decreased expression of the secretory proteins, serum paroxygenase, serum albumin, and peroxide-reducing protein IV. The correlation of these protein changes with the clinical and histological data and their occurrence before the onset of biochemical changes suggest that they can serve as predictive biomarkers of compounds with a propensity to induce liver steatosis [69]. The study used rats as a model animal. Although rats are a commonly used model in toxicology studies, they differ from human liver metabolism in several manners. Proteomic changes in the rat model do not necessarily fully reflect physiological responses in humans, which limit the clinical applicability of the results. However, with the continuous improvement in sensitivity of mass spectrometers and progress of technologies such as click chemistry, proteomics analysis can better explain the phenomenon of the discovered drug toxicity response in the clinic and become more comprehensive and precise in the evaluation of drug off-target effects. APAP is metabolized in excess to strongly reactive NAPQI, which triggers liver damage via the covalent modification of proteins. Proteomics detected proteins associated with oxidative stress, such as GATM, PARK 7, PRDX 6, and VDAC 2 modified by NAPQI, to explain the mechanism of APAP toxicity [70]. Active probe-based proteomic analysis (activity-cased protein profiling, ABPP) can effectively identify the targets of these off-target effects to reduce the risk of toxicity and optimize the safety assessment of drugs [71].

### 4.4. Dynamic Follow-Up and Monitoring

Proteomics technology provides a powerful tool for the dynamic follow-up and monitoring of patients. By regularly analyzing changes in protein expression, one can better understand the disease progression, treatment response, and relapse risk to offer crucial support for personalized treatment and early intervention. In kidney transplant patients, urinary proteomics have proven to be an effective method for the early identification of biomarkers associated with acute rejection (AR). Researchers have conducted quantitative analyses of proteins in the urine of post-transplant patients and successfully identified proteins that are closely related to AR. For example, Sigdel et al. [72] used iTRAQ-based proteomics and validated urinary protein biomarkers in 262 biopsy-confirmed kidney transplant recipients through targeted ELISA. The analysis revealed that 9 of 69 different urinary proteins (including HLA-DRB1, FGG, FGB, FGA, KRT14, HIST1H4B, ACTB, KRT7, and DPP4) were significantly associated with the stable graft function, BK virus nephropathy, and chronic allograft injury (*p* < 0.01; fold change > 1.5). Among these, three proteins (FBG, FGG, and HLA-DRB1) were identified as candidate biomarkers for acute rejection. Additionally, in 2018, Lim et al. [73] conducted a proteomic analysis on collected urine samples to identify candidate biomarkers in urinary exosomes related to acute T-cell-mediated rejection (TCMR). The study found that 17 proteins were upregulated in TCMR patients, and the Western blot analysis confirmed the proteomic expression levels of five candidate biomarkers for each patient. Among all candidate biomarkers, Tetraspanin-1 and Hemopexin were significantly elevated in TCMR patients, which indicates their strong associations with rejection. This dynamic monitoring approach based on proteomics provides strong support for personalized treatment in kidney transplant patients. Compared with traditional serum creatinine testing and kidney biopsies, urinary proteomics offers a non-invasive and highly sensitive detection method to help doctors better predict and manage rejection and ensure the long-term survival of the transplanted kidney.

## 5. Challenge

Despite the great potential of proteomics in biomedical research, many challenges remain in its practical application.

Proteomic analysis often requires complex sample processing steps, including the extraction, isolation, and purification of proteins [74]. For complex biological samples (such as tissues and body fluids), effective separation and quantification of low-abundance proteins remain challenging. Especially when detecting low-abundance proteins, the MS analysis demands the extreme sensitivity of the device, which makes this process more complex. Moreover, high-precision mass spectrometers are expensive to equipment and maintain and operate, which partly limits the spread of proteomics in a wide range of clinical applications. The massive amount of data of proteome students contains thousands of proteins and their post-translational modification forms, and how to extract meaningful biological information from them is a key issue. The integration of data requires efficient bioinformatics tools and algorithms, but existing tools and databases often fail to comprehensively cover all protein modifications and isoforms, which limits the interpretation and application of proteomics data.

Protein concentrations in biological samples are in an extremely wide range from high-abundance proteins to low-abundance biomarkers, where concentration differences may differ by several orders of magnitude [75]. Low-abundance proteins (such as hormones and transcription factors) are often difficult to detect, whereas high-abundance proteins (such as albumin in plasma) can mask the signals of low-abundance proteins and increase the difficulty of identification and quantification. Post-translational modifications of proteins (e.g., phosphorylation, acetylation, glycosylation, etc.) increase the diversity of their function and localization, and the complexity of these modifications makes the identification and quantification of proteins in different modified forms another challenge for proteomics.

In the context of translational medicine, proteomics has great potential, but its research findings are difficult to apply to clinical practice. In the past, monitoring plasma with protein biomarkers to reflect the physiological status of individuals has been an important goal of protein science. However, despite recent progress in mass spectrometry in plasma proteome analysis, the complexity of the plasma proteome, its inherent variability in the population, and the prevalence of influencing factors such as age, sex, and lifestyle, make this approach extremely challenging [10]. Therefore, the development of simple, economical and efficient detection platforms for large-scale clinical applications continues to be a key issue facing proteomics.

As an important tool to understand biological systems, proteomics commonly requires a combination with other omics techniques such as genomics, transcriptomics, and metabolomics to provide a comprehensive resolution of complex biological processes. However, integrating these multi-omics data and performing efficient analyses face technical and analytical challenges. Comparing static genomics data with dynamic transcriptomic, proteomic and metabolomic data is a challenge. Moreover, the lack of unified standards in experimental design, data generation, and analysis processes further aggravates the complexity of data integration. The heterogeneity of the data and batch effects (batch effects) can affect the accuracy and comparability of the analysis results, and they must be adjusted and corrected by complex statistical methods to ensure the reliability of the results.

## 6. Future Prospects

### 6.1. Emerging New Technologies

In recent years, remarkable advances in proteomics technology have greatly promoted research in the fields of life sciences and medicine. In particular, the development of new technologies, such as high-throughput mass spectrometry, protein chips, and single-cell proteomics, has enabled us to penetrate the proteome at higher resolutions and scales to provide more powerful tools for disease research and precision medicine.

The rapid development of high-throughput mass spectrometry has enabled the simultaneous identification and quantification of thousands of proteins, which greatly improves the efficiency and accuracy of proteomics studies. This technology is widely used in the discovery of disease markers, screening of drug targets [76], and validation of biomarkers. One core technology of high-throughput mass spectrometry is tandem mass spectrometry (MS/MS). Through multiple rounds of mass spectrometry analysis, the target molecules are first selected; then, the fragments are detected twice to obtain their structural information, which greatly improves the accuracy and resolution of detection [77]. Moreover, the introduction of automated sample preparation and sample injection systems enables high-throughput mass spectrometry to process large numbers of samples in a short time, reduce human operation errors and significantly improves the flux. For example, a liquid chromatography-mass spectrometry (LC–MS) system can be used for sample separation and injection. High-throughput mass spectrometry also usually uses high-resolution mass spectrometers (such as Orbitrap or time-of-flight mass spectrometers), which can simultaneously detect many ions and ensure high-precision data in a short time [78].

Protein chip technology is a high-throughput and high-sensitivity protein detection method. It enables efficient detection and analysis by immobilizing many proteins on solid-phase supports for specific interactions with the target proteins in the sample to be tested. A study developed a microarray-based method for microproteins, which was successfully used to identify novel inhibitors of heat shock proteins (HSPs) using only 300 pmol of protein for analysis, which reduced sample waste [79]. This study demonstrates the potential of microarray technology for use in drug development, particularly in the development of therapeutic agents for HSPs, and highlights the importance of this technology in accelerating the screening of therapeutic agents.

Before the emergence of single-cell proteomics (SCP) technology, traditional proteomics methods relied on large-scale cell or tissue samples for analysis, which easily masked the heterogeneity among individual cells and failed to identify the protein expression differences in different cell types. Single-cell proteomics can capture the protein expression diversity in different cell populations and reveal the cellular functional differences that may be overlooked in traditional proteomics. In the past 5–6 years, with the rise and rapid development of single-cell proteomics, many innovations and applications have emerged. To increase the sensitivity while increasing the flux, isobaric labelling strategies (e.g., TMT) have been introduced in the SCP field for sample multiplexing [80]. In 2018, the first SCoPE-ms workflow for the successful implementation of this strategy was developed and provided strong support for single-cell proteomics (SCP) [81].

SCPs provide a completely new perspective to resolve the biological complexity at the cell level. Schoof et al. [82] used the SCoPE workflow to characterize the cellular hierarchy in acute myeloid leukaemia (AML), which represents a significant advance in SCP technology. This study is particularly important because it validates the accuracy and reliability of the SCP in analyzing cell differentiation in complex systems. Compared with FACS data, SCP can serve as a complementary tool to characterize haematological disease cell populations and may even become an alternative tool to provide new insights into leukaemia progression and classification.

Orsburn et al. [83] further promoted the application of single-cell proteomics to study post-translational modifications (PTMs) at single-cell resolution. Using an SCoPE2 workflow and a high-throughput timsTOF mass spectrometer, the research team analyzed over 40,000 tandem MS data points in 30 min. One of the most striking findings was the detection of multiple PTM types, including phosphorylation, acetylation, methylation, dimethylation, succinylation, hydroxybutyylation, crotonylation, and cysteine trioxidation. This was the first time that multiple types of PTMs were successfully identified in single cells, which demonstrated that SCPs could provide high-resolution insight into cellular mechanisms at the level of protein modification [83].

Despite significant progress in SCP, challenges remain in developing high-throughput, automated, and scalable technologies. Due to the small number of proteins in single cells, detecting low-abundance proteins remains challenging at the nak to picak level. Despite the high sensitivity of modern mass spectrometry technologies (e.g., Orbitrap and TOF-MS), the detection of low-abundance proteins remains a bottleneck in single-cell analysis, especially those that play a key role in cell signaling and regulation.

In the future, as the technology becomes more mature, single-cell proteomics may focus more on data integration to combine data from genomics and transcriptomics to comprehensively resolve the multidimensional biological characteristics of individual cells. Furthermore, with improved instrument sensitivity and optimization of analytical algorithms, SCP is expected to play a greater role in a wider range of clinical and basic research scenarios.

### 6.2. Integrative Approaches

Proteomics has important applications in protein expression and function studies, but its limitations cannot be ignored. It cannot cover information at the gene, transcription, metabolism and other levels and faces challenges such as low-abundance protein detection, complexity of post-translational modification, and difficulty in data processing. Therefore, proteomics must usually be combined with techniques such as genomics, transcriptomics, and metabolomics to more comprehensively reveal the complexity and functional regulatory mechanisms of biological systems. In 2014, Robert et al. [79] published an article titled “Linking cancer genome to proteome: NCI’s investment into proteogenomics” to discuss the National Cancer Institute (NCI) strategic initiatives to integrate genomic and proteomics data through proteomics to enhance cancer research and treatment. This work focused on how proteomics bridged the gap between genetic mutations and their functional protein-level consequences to gain a deeper understanding of cancer biology.

The integration of multidimensional omics technologies also helps better understand the mechanisms of disease development and progression, especially in neurodegenerative diseases. A research team combined transcriptome and proteome data from the Drosophila tau disease model [84], revealed significant age- and tau-dependent changes in gene and protein expression, and detected the overlap between gene expression profiles in tau and the human brain. This work emphasized the importance of considering the transcriptome and proteome changes to fully understand the complexity of tau disease and its progression during ageing. In addition, integrating transcriptome and proteomic data from placental tissue helped identify key molecular changes that caused excessive migration in trophoblast cells, which are at the heart of placental implantation (PA) pathology [85]. The work revealed 37,743 differentially expressed transcripts and 160 proteins between PA patients and controls, although the overlap between the transcriptomic and proteome data was small (only 0.09%). This finding suggests the existence of complex post-transcriptional regulatory mechanisms. Among the differentially expressed proteins, methyl-CpG binding domain protein 2 (MeCP 2), podocyte protein (PODN) and apolipoprotein D (ApoD), which are involved in the inhibition of cell migration, were significantly downregulated in PA patients. This downregulation promotes excessive trophoblast cell migration and causes the pathological over-invasion observed in PAs.

With the increasing popularity of AI, the combination of proteomics and artificial intelligence (AI) has opened new possibilities for biomedical research, disease diagnosis, and drug development in multiple fields. Proteomics studies generate such large and complex data, which AI, especially machine learning (ML) technology, can help parse [86]. The development of the AlphaFold database represents a major breakthrough of AI in proteomics [87]. It uses a deep learning algorithm to predict the 3D structure of proteins with much higher accuracy than previous prediction methods. Artificial intelligence in protein is the most important achievement; the Mann group of Rosenberger et al. [88] described a new technology called single-cell depth visual proteomics (scDVP) and how to use it from the cell slices of the mouse liver cell environment-dependent space proteome. They combined submicron-resolution imaging, artificial intelligence (AI)-based image-based single-cell phenotypic analysis and separation with ultrasensitive proteomics workflows to study protein abundance in single cells while retaining their spatial background (Figure 4). This approach enables researchers to link protein profiles to complex cellular phenotypes at the single-cell level and provide insights into cellular heterogeneity in diseases such as cancer.

These examples indicate that the combination of proteomics and other omics technologies with AI provides us with a more comprehensive understanding of the complexity of biological systems.

## 7. Conclusions

The application of proteomics in precision medicine provides a new breakthrough for disease management and personalized treatment. Through in-depth analyses of patient protein expression profiles, post-translational modifications and their dynamic changes, proteomics can help identify biomarkers associated with a specific disease. These markers can be used for early diagnosis and to predict disease progression and assess response to treatment. Moreover, proteomics can reveal drug targets and off-target effects to optimize the drug design and therapeutic strategies to reduce side effects. Through the integration with multi-omics data such as genomics and transcripomics, proteomics makes precision medicine more comprehensive, promotes the development of personalized treatment programs, and makes the treatment more targeted at specific pathological characteristics of patients to improve their efficacy and prognosis. In the future, with continuous technological advances and increased interdisciplinary cooperation, proteomics is expected to play a more important role in the biomedical field. Moreover, we must focus on the challenges such as technical issues and actively seek solutions to promote the further development of proteomics research.

## Figures and Tables

**Figure 1 biomedicines-13-00681-f001:**
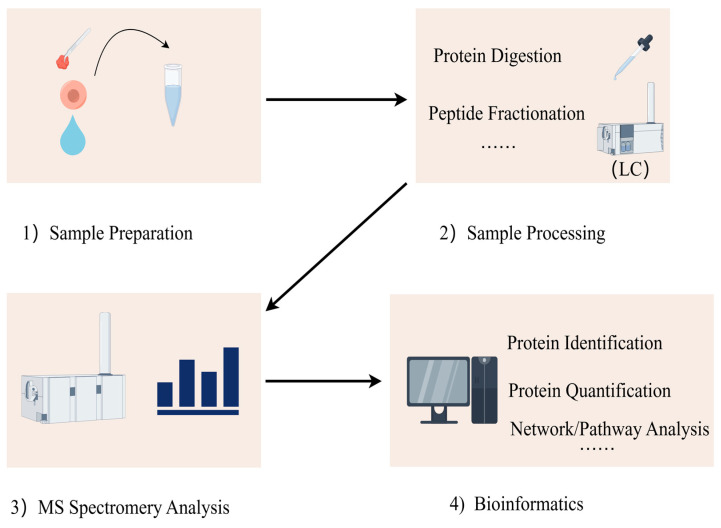
Proteins are extracted from various biological samples, including tissues, cells, and body fluids. Proteins are digested with proteolytic enzymes (i.e., trypsin) for easier subsequent mass spectrometry detection. Liquid chromatography (LC) is used to separate the digested peptides. Mass spectrometry can be used to identify the masses of peptides and their primary structures. Mass spectrometry data are aligned with protein databases to identify proteins and their peptide sequences. Bioinformatics tools can be used to functionally annotate the identified proteins, analyze the signaling pathways, analyze the protein interactions, and study the post-translational modifications of proteins.

**Figure 2 biomedicines-13-00681-f002:**
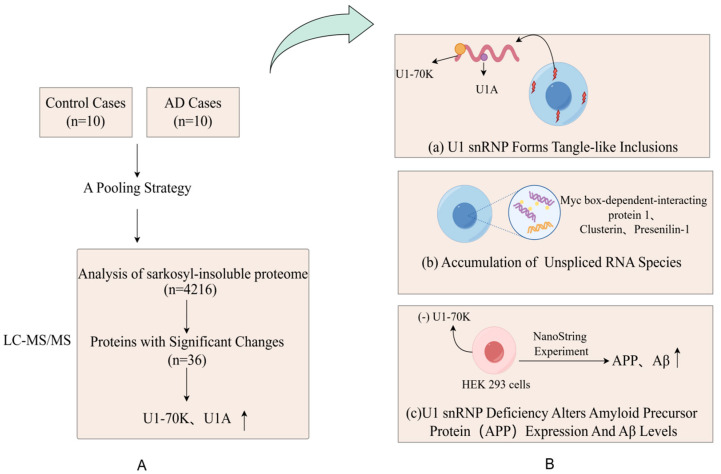
(**A**) Proteomic comparison reveals that U1-70K and U1A are enriched in the sarkosyl-insoluble proteome of AD patients. They designed a pooling strategy with replicates to simplify the analysis of protein aggregates in the cortical tissues harvested from 10 AD patients and 10 age-matched, nondemented patients (**B**) (**a**) The pathological examination revealed significant and widespread extranuclear aggregation of U1snRNP components in the neuronal cells of AD patients; (**b**) Researchers revealed widespread changes in RNA processing in the brains of AD patients, including the accumulation of unspliced RNA species, by comparing RNA in the brains of AD and control groups; (**c**) Through the knockdown of U7-70K, researchers found an increase in levels of amyloid precursor protein, which indicates that the abnormalities in U1snRNP may be associated with the pathological process of AD.

**Figure 3 biomedicines-13-00681-f003:**
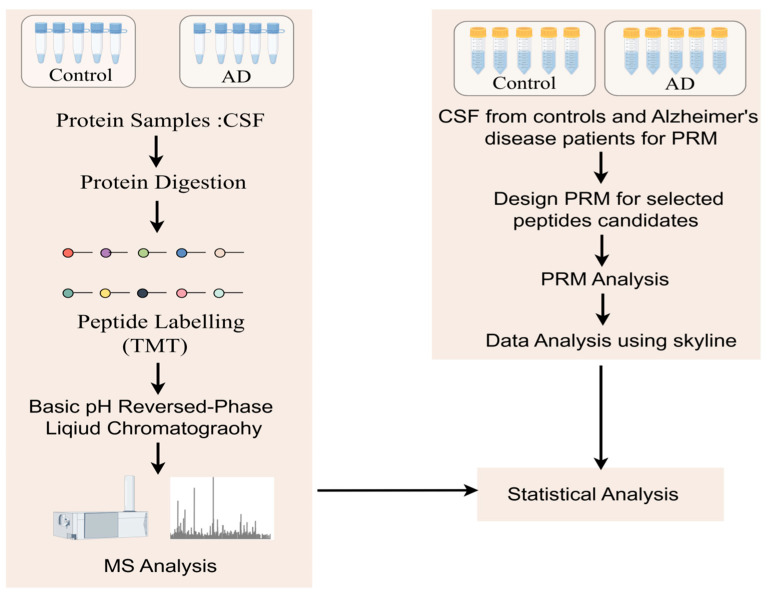
Workflow to study protein changes in the cerebrospinal fluid (CSF) of Alzheimer’s disease (AD) patients: A MARS-14 immunoaffinity depletion column was used to remove high-abundance proteins from CSF samples. In the discovery step, a TMT-based quantitative proteomics method was used to compare the CSF proteome of cognitively normal individuals with that of AD patients. For the subset of molecules that show significant changes in AD, parallel reaction monitoring (PRM) assays were used for validation.

**Figure 4 biomedicines-13-00681-f004:**
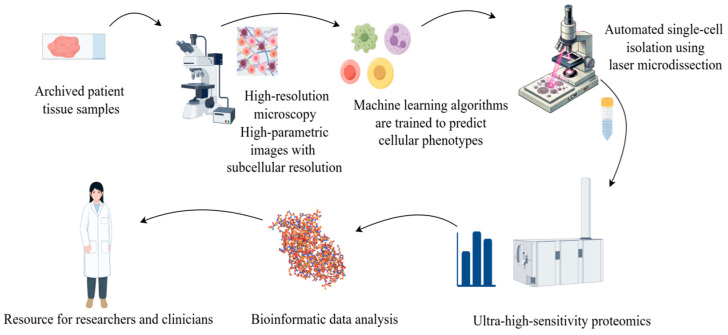
Using microscopy techniques, it is possible to observe the structure of tissue or cellular samples. This process captures the morphological characteristics of cells and provides visual information about the cell’s state or marker expression through specific staining or fluorescent labelling. The obtained cell images are analyzed and classified using machine learning or artificial intelligence algorithms. During the image analysis phase, features of each single cell (such as morphology, size, and staining intensity) are extracted, and based on these features, the cells are classified or grouped. Based on the image classification, each individual cell is subjected to Laser Capture Microdissection (LCM) and mass spectrometry analysis. Mass spectrometry enables the precise determination of the proteome data of each cell, including the types and relative abundances of proteins. The mass spectrometry data are integrated with the image classification information, and through bioinformatics methods, the functional state, cell type identity, and heterogeneity among the cell populations are revealed.

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
