# Peer review of "Application of the Human Proteome in Disease, Diagnosis, and Translation into Precision Medicine: Current Status and Future Prospects"

_biomedicines, 2025, doi:10.3390/biomedicines13030681_

Round 1

Reviewer 1 Report

Comments and Suggestions for Authors

The review entitled “Application of the Human Proteome in Disease, Diagnosis, and Translation into Precision Medicine: Current Status and Future Prospects” and submitted in Biomedicines is well written and summarizes the existing studies of human proteomics technology in the medical field with a focus on the development mechanism of a disease and its potential in discovering biomarkers. This review is enough comprehensive, the appropriate references are used. I believe this manuscript can be published in the current form.

Author Response

Thank you very much for your thoughtful review of my manuscript titled “Application of the Human Proteome in Disease, Diagnosis, and Translation into Precision Medicine: Current Status and Future Prospects。”(biomedicines-3406913). I deeply appreciate your time and expertise in evaluating this work.

Your encouraging comments have been both motivating and insightful. Such recognition from a scholar in this field is a tremendous encouragement to me.

Once again, thank you for your support in advancing this study.

Reviewer 2 Report

Comments and Suggestions for Authors

The authors discuss significant advantages and application scenarios of proteomics technology in disease diagnosis, drug development, and personalized treatment.

The review is well-written.

The clinical applications of proteomics with emphasis on premalignant lesions especially bladder cancer) squamous cell carcinoma) should be elaborated.

The authors could describe alterations in protein expression in dilated cardiomyopathy. Specific proteins that are hyperubiquitinated in diseased hearts could be discussed.

Peptide fingerprints can be generated from various body fluids. The authors can discuss this fact.

Proteomics-based technologies being used as candidates for vaccine production should be added.

The progress of a protein microarray enough to explore the function of a complete genome is challenging. What may be the challenges?

The diverse proteomics approaches such as mass spectrometry (MS) have been developed to analyze complex protein mixtures with higher sensitivity. These facts could be elaborated.

Chromatography-based techniques, such as ion exchange chromatography, are versatile tools for purifying proteins based on the charged groups on their surface. The authors can discuss these facts.

Protein microarrays can be classified into three categories; analytical protein microarray, functional protein microarray and reverse-phase protein microarray nd all these categories should be discussed.

Author Response

Thank you for your valuable feedback on our manuscript, and I will provide a comprehensive point-by-point response to your comments.

1、The authors could describe alterations in protein expression in dilated cardiomyopathy. Specific proteins that are hyperubiquitinated in diseased hearts could be discussed.

ResponseThankyou for your valuable suggestion!

We have revised the manuscript in your opinion and added relevant sections, which are added to "3.2. Discovery of Biomarkers" and "3.3.2. Changes in Protein Modifications"

A research team employed proteomics techniques, specifically mass spectrometry and label-free quantification (LFQ), to analyze plasma samples from patients with dilated cardiomyopathy (DCM) in order to identify potential biomarkers associated with the disease, particularly those that could predict treatment outcomes. After categorizing DCM patients based on their treatment response (LVRR+ and LVRR−), 45 differentially expressed proteins were identified. Among them, fructose-1,6-bisphosphate aldolase (ALDOB) emerged as a promising biomarker, as its levels were elevated in patients with poor treatment response, potentially indicating a worse prognosis[46].(page 7)

 Researchers have investigated the phosphorylation and degradation mechanisms of αB-crystallin (CryAB) during Coxsackievirus B3 (CVB3) infection using proteomics techniques, particularly mass spectrometry, Western blotting, phosphorylation labeling, and protein degradation assays. The study found that CryAB undergoes phosphorylation at specific serine residues following CVB3 infection, and the phosphorylated CryAB is subsequently degraded via the ubiquitin-proteasome pathway. This degradation leads to the disruption of the myocardial cytoskeletal network, further promoting viral replication and the progression of cardiomyopathy. These findings reveal the molecular mechanisms underlying viral myocarditis and provide potential therapeutic targets for future treatments[51].(page 8)

 References:

(1).Klimentova J, Rehulka P, Stulik J, Vozandychova V, Rehulkova H, Jurcova I, Lazarova M, Aiglova R, Dokoupil J, Hrecko J, Pudil R. Proteomic Profiling of Dilated Cardiomyopathy Plasma Samples ─ Searching for Biomarkers with Potential to Predict the Outcome of Therapy. J Proteome Res. 2024 Mar 1;23(3):971-984. doi: 10.1021/acs.jproteome.3c00691. Epub 2024 Feb 16. PMID: 38363107; PMCID: PMC10913098.

(2)Fung G, Wong J, Berhe F, Mohamud Y, Xue YC, Luo H. Phosphorylation and degradation of αB-crystallin during enterovirus infection facilitates viral replication and induces viral pathogenesis. Oncotarget. 2017 Aug 19;8(43):74767-74780. doi: 10.18632/oncotarget.20366. PMID: 29088822; PMCID: PMC5650377.

2、Proteomics-based technologies being used as candidates for vaccine production should be added.

ResponseThankyou for your valuable suggestion!

We revised the manuscript in your opinion and added relevant parts, which were increased to "4.5 Vaccine production"

Proteomics is increasingly playing a crucial role in vaccine development. By systematically analyzing the protein composition, expression levels, and interactions of pathogens or host cells, proteomics offers novel perspectives and strategies for vaccine research. Unlike traditional vaccine development, which relies on limited knowledge of pathogens and empirical approaches, proteomics enables a comprehensive analysis of the pathogen proteome, allowing for the precise identification of potential antigenic targets and immune-modulatory molecules. For example, through the integration of proteomics and immunoinformatics, researchers have designed a multi-epitope vaccine targeting multidrug-resistant Enterococcus faecium. Proteomics allowed the prediction and selection of immune epitopes from the target protein PBP 5, while molecular dynamics simulations and molecular docking were used to evaluate the vaccine's stability and immune activation potential. Additionally, codon optimization and structural refinement were employed to enhance the vaccine’s stability and expression efficiency. With ongoing advancements in technology, proteomics is expected to play an even more significant role in personalized vaccine development, novel vaccine delivery system design, and immune response mechanism studies, heralding a revolutionary shift in vaccine production[76].(page 12)

 Reference

Dey J, Mahapatra SR, Raj TK, Kaur T, Jain P, Tiwari A, Patro S, Misra N, Suar M. Designing a novel multi-epitope vaccine to evoke a robust immune response against pathogenic multidrug-resistant Enterococcus faecium bacterium. Gut Pathog. 2022 May 27;14(1):21. doi: 10.1186/s13099-022-00495-z. PMID: 35624464; PMCID: PMC9137449.

3、Chromatography-based techniques, such as ion exchange chromatography, are versatile tools for purifying proteins based on the charged groups on their surface. The authors can discuss these facts.

ResponseThankyou for your valuable suggestion!

We revised the manuscript in your opinion and added relevant parts, which were added to "6.1. Emerging New Technologies"

Ion exchange chromatography (IEC) is a separation technique that isolates molecules based on differences in their electrostatic properties. This technique offers several advantages, including high sample throughput, broad applicability (especially for proteins and enzymes), moderate cost, high resolution, the ability to perform quantitative analysis, and ease of scaling and automation. As a result, it has become one of the most widely used methods in all liquid chromatography (LC) techniques [82]. By precisely adjusting the pH gradient, researchers can control the migration of individual components through the chromatographic column, thereby enhancing separation selectivity. This approach is particularly effective in separating charge variants. For instance, in the separation of monoclonal antibodies, it can efficiently distinguish between different charge isoforms, such as glycosylation variants. Moreover, by integrating IEX with techniques such as mass spectrometry (MS), multidimensional chromatography, ion mobility spectrometry (IMS), and multi-angle light scattering (MALS), IEX can provide more accurate and comprehensive information. This integration enables in-depth analysis of complex biological samples, especially in the study of protein charge variants, glycosylation modifications, and aggregates, thereby demonstrating its strong potential in various research applications [83].(page 14)

 References:

(1).Wallace RG, Rochfort KD. Ion-Exchange Chromatography: Basic Principles and Application. Methods Mol Biol. 2023;2699:161-177. doi: 10.1007/978-1-0716-3362-5_9. PMID: 37646998.

(2).Imiołek M, Fekete S, Rudaz S, Guillarme D. Ion exchange chromatography of biotherapeutics: Fundamental principles and advanced approaches. J Chromatogr A. 2025 Feb 8;1742:465672. doi: 10.1016/j.chroma.2025.465672. Epub 2025 Jan 9. PMID: 39805233.

4、Protein microarrays can be classified into three categories; analytical protein microarray, functional protein microarray and reverse-phase protein microarray nd all these categories should be discussed.

ResponseThankyou for your valuable suggestion!

We revised the manuscript in your opinion and added relevant parts, which were added to "6.1. Emerging New Technologies"

Protein microarrays can currently be categorized into three types: analytical protein microarrays, functional protein microarrays, and reverse-phase protein microarrays (RPPA). Functional protein microarrays are not only used to study the fundamental properties of proteins but are also widely employed in screening novel drug candidates and drug targets, making them invaluable in the drug discovery process. Moreover, functional protein microarrays facilitate a deeper understanding of how proteins function both inside and outside of cells, and the specific mechanisms underlying their interactions, playing a crucial role in the comprehensive analysis of protein functions. Analytical protein microarrays are primarily used for monitoring protein expression. For instance, in biomarker screening, antibody microarrays efficiently detect and quantify specific proteins or antigens in biological samples. This technology is especially suited for high-throughput analysis of complex samples and is widely applied in clinical diagnostics and environmental monitoring. Through efficient capture and quantitative analysis, analytical protein microarrays have found broad applications in modern biotechnology [85]. Reverse-phase protein microarrays, on the other hand, allow for efficient and sensitive analysis of proteins in tumor samples, particularly protein expression and modifications. This technology employs microarray techniques to simultaneously analyze multiple proteins through dot spotting, with quantitative data obtained via antibody detection and signal enhancement. Reverse-phase protein microarrays hold significant potential in precision oncology, particularly in cancer molecular subtyping, drug response prediction, and personalized cancer treatment. By combining with other biotechnologies, RPPA offers more detailed and in-depth information for precision therapies and is poised to become an essential tool in personalized cancer treatment [86].(page 14)

 Reference

(1).Yang L, Guo S, Li Y, Zhou S, Tao S. Protein microarrays for systems biology. Acta Biochim Biophys Sin (Shanghai). 2011 Mar;43(3):161-71. doi: 10.1093/abbs/gmq127. Epub 2011 Jan 21. PMID: 21257623; PMCID: PMC7117600.

(2).Masuda M, Nakagawa R, Kondo T. Harnessing the potential of reverse-phase protein array technology: Advancing precision oncology strategies. Cancer Sci. 2024 May;115(5):1378-1387. doi: 10.1111/cas.16123. Epub 2024 Feb 26. PMID: 38409909; PMCID: PMC11093203.

As for the other comments, we thank you very much for these comments, but due to the little literature I found, I failed to make the revisions. Thank you again for your valuable comments!

Round 2

Reviewer 2 Report

Comments and Suggestions for Authors

All necessary corrections were done.